# Exploring the preservation of a parasitic trace in decapod crustaceans using finite elements analysis

**Nathan L. Wright** [1]*, **Adiël A. Klompmaker**[2], **Elizabeth Petsios**[1]

**1** Department of Geosciences, Baylor University, Waco, Texas, United States of America, **2** Department of Museum Research and Collections & Alabama Museum of Natural History, University of Alabama, Tuscaloosa, Alabama, United States of America

* Nathan_wright1@baylor.edu

**Data Availability Statement:** All analysis von Mises stress results is available through the repository Dryad at the following DOI/URL: https://doi.org/10.5061/dryad.8cz8w9gz4.

## Abstract

The fossil record of parasitism is poorly understood, due largely to the scarcity of strong fossil evidence of parasites. Understanding the preservation potential for fossil parasitic evidence is critical to contextualizing the fossil record of parasitism. Here, we present the first use of X-ray computed tomography (CT) scanning and finite elements analysis (FEA) to analyze the impact of a parasite-induced fossil trace on host preservation. Four fossil and three modern decapod crustacean specimens with branchial swellings attributed to an epicaridean isopod parasite were CT scanned and examined with FEA to assess differences in the magnitude and distribution of stress between normal and swollen branchial chambers. The results of the FEA show highly localized stress peaks in reaction to point forces, with higher peak stress on the swollen branchial chamber for nearly all specimens and different forces applied, suggesting a possible shape-related decrease in the preservation potential of these parasitic swellings. Broader application of these methods as well as advances in the application of 3D data analysis in paleontology are critical to understanding the fossil record of parasitism and other poorly represented fossil groups.

## Introduction

Parasitism is the most common and ecologically impactful mode of life today, as parasitic taxa comprise at least 40% of described extant species and are critical to ecosystem trophic structure [1]. However, evidence of parasitic behavior in the fossil record is uncommon [2]. Studies describing cases of potential fossil evidence of parasites span several decades [3–5], but broader synthesis and interpretation of fossil parasitic evidence has only recently been undertaken because of broader data access, an increase in studies, and computationally intensive tools [6]. Despite these advances, existing evidence and data of fossil parasitism is limited, taxonomically biased, and often over-represented by sites of exceptional preservation. Taphonomic bias is suggested to be a main factor causing these disparities because many ubiquitous modern parasitic taxa are soft-bodied, small, and endoparasitic, with little chance of producing recognizable fossil evidence [7, 8]. Many modern marine parasites create distinct traces on their host's

**Funding:** The author(s) received no specific funding for this work.

**Competing interests:** The authors have declared that no competing interests exist.

skeletal elements, yet, for many taxa, fossil evidence of these parasites and traces is uncommon. Biases against evidence of parasitism in the fossil record have yet to be critically examined or quantified [9–13]. As a result of the scarcity of parasitic evidence, the fossil record of parasitism remains a major gap in the scientific understanding of the evolutionary and ecological history of life on earth.

Crustaceans are a highly diverse, abundant, and globally distributed group of arthropods, with a fossil record spanning more than five hundred million years. Crustaceans are well documented as a group in which many novel parasitic interactions have arisen independently several times, with crustaceans exhibiting great diversity as both hosts and parasites [12]. Some of the parasitic relationships among crustaceans produce distinct, preservable traces, which can be identified from fossil host specimens [12]. These features make crustaceans an excellent model group for studying the fossil record of parasitism. The fossil record of crustaceans is limited, and often fragmentary, owing to a lightly to moderately (and often heterogeneously) calcified exoskeleton that readily disarticulates in the absence of soft tissue [14]. Many crustaceans are found in concretions, typically calcite or siderite, which preserve high fidelity 3D fossils, sometimes with original exoskeletal material remaining [15–17]. Fossiliferous concretions are a unique and exceptional taphonomic scenario, suggesting rapid burial and rapid microbially and geochemically mediated concretionary growth of carbonate minerals, which only occurs under specific conditions [18]. Crustaceans have also been the subject of taphonomic study, revealing the fragmentary and biased nature of the crustacean fossil record, controlled by differences in calcification across individuals and between taxa and depositional conditions [9, 14, 19].

Extant crustacean hosts and their parasites are the subject of much study, as a result of being ubiquitous in marine ecosystems, as well as their significant economic and ecological importance, but fossil evidence of parasitism among crustaceans is limited [12]. Suboval branchial swellings found on decapod carapace fossils, identified as the ichnotaxon *Kanthyloma crusta* [11], are among the most well understood examples of fossil parasitism of crustaceans. These traces are inferred to be induced by parasitic epicaridean isopods that commonly produce identical traces on modern decapods [11]. These distinct fossil swellings have been found on diverse decapod fossils since an early peak in the number of parasitized host species in the Late Jurassic, found predominantly in Europe [20]. Likely using the Tethys as a dispersal pathway, *K. crusta* has been observed on decapod fossils globally through the Cretaceous and Cenozoic, although with lower host diversity than seen in the Late Jurassic [21]. Extant isopods in the family Bopyridae, which induce branchial swellings attributable to *K. crusta*, are globally widespread and infest diverse decapod species [22], in contrast to the relatively poor Cenozoic fossil record of *K. crusta*. Modern bopyrid isopods are widespread parasites of decapods, typically at relatively low prevalence (<5%) within host populations, but can instigate rapid host population collapse when introduced to a naïve host population [13, 23].

As high-quality specimens of fossil decapods are rare and scientifically valuable, particularly those bearing additional abnormalities such as traces of parasitism, destructive sampling techniques and intrusive attempts to reveal fossils from the matrix are inadvisable. CT scanning has seen a dramatic increase in widespread use for paleontological study over the last 20 years, as the technology has become more accessible, and CT data has become easier to analyze and manipulate [24–30].

Computational simulation techniques, such as finite element methods and computational fluid dynamics, have been used to simulate physical forces acting on 3D models to explore functional morphology and ecology in fossils [31, 32]. Studies over the last two decades have applied finite elements analysis (FEA) to paleontological and zoological material, including morphological study of fossil arthropods and modern crustaceans [32, 33], and found that

high-fidelity finite-element models of living and fossil organisms are robust at modelling strain and resolving relationships between form and function [34–36].

The aim of this study is to present the first application of FEA in evaluating the impact of the parasite-induced trace fossil *Kanthyloma crusta* on host preservation by exploring how deformations of carapace shape alter the stress response of the carapace to external forces, and therefore preservation potential of the deformation and the host. Specimens of modern and fossil decapod crustaceans with branchial swellings attributed to isopod infestation (*K. crusta*) were CT-scanned, and FEA was conducted to observe differences in the magnitude and distribution of stress on healthy and swollen branchial chambers.

## Methods

### Specimens studied

Seven specimens of recent and fossil decapod crustacean specimens with swellings attributable to *K. crusta* were studied through institutional loans: three Recent specimens of *Munida valida* [37] from the Gulf of Mexico preserved in 70% ethanol (Fig 1A–1C), three complete fossil specimens of *Macroacaena rosenkrantzi* [38] in siderite concretions from the Maastrichtian of Greenland, from a site referred to as the oyster-ammonite locality (Fig 1D–1F), and one isolated fossil carapace of *Panopeus nanus* [39] in lower Miocene limestone from the Duncans Quarry, Trelawny Parish, Jamaica (FLMNH-IP locality XJ015) (Fig 1G). These specimens were studied through institutional loans from Texas A&M University in College Station, Texas, USA (TAMU), the Natural History Museum of Denmark in Copenhagen, Denmark (NHMD) stored in the Type and Illustrated Paleontology Collection, and the University of Florida, Florida Museum of Natural History, Invertebrate Paleontology in Gainesville, Florida, USA (UF-FLMNH-IP), respectively. The recent specimens of *Munida valida* are corpses, the fossil specimens of *Macroacaena rosenkrantzi* likely represent corpses, and the fossil specimen of *Panopeus nanus* likely represents a molt. Each specimen has one swollen branchial chamber, in either the left (n = 3), or right (n = 4) chamber. No permits were required for the described study, which complied with all relevant regulations. We used the software Polycam [40] to create 3D surface models of the specimens from several photographs for additional visualization of the specimens (Fig 1D–1F), in addition to digital photography (Fig 1A–1C and 1G).

### Scanning and CT data preparation

X-ray computed tomography (CT) is a technology that uses an X-ray source to construct several cross-sectional images, which can then be combined and processed to form a 3D volumetric render of the scanned subject [24]. Specimen 3D data for analysis was captured using a North Star Imaging (NSI) X3000 industrial CT x-ray inspection system. All CT scans were conducted at sub-35μm voxel resolution, using a voltage of 80 kV, current of 480μA, with the specimen platform rotating at 4.5 degrees per second, a detector framerate of 12.5 fps, an average source to specimen distance of 128 mm, and a source to detector distance of 539 mm. Simultaneous CT scans of multiple specimens (including the *Munida* and *Macroacaena* specimens) utilized the NSI SubpiX [41] scanning technique, which combines multiple consecutive scans at sub-pixel offsets for increased resolution. CT data was initially processed and visualized using the NSI efX software and was then exported as stacks of TIFF images for further preparation. The image stacks were imported into the software Dragonfly ORS 2022.1 [42] for visualization and 3D conversion of the CT data. The specimen data was segmented from the raw image slices using a mixture of manual segmentation using Dragonfly's ROI (region of interest) painter tools and manually trained machine learning segmentation models created in Dragonfly's Segmentation Wizard tool [42].

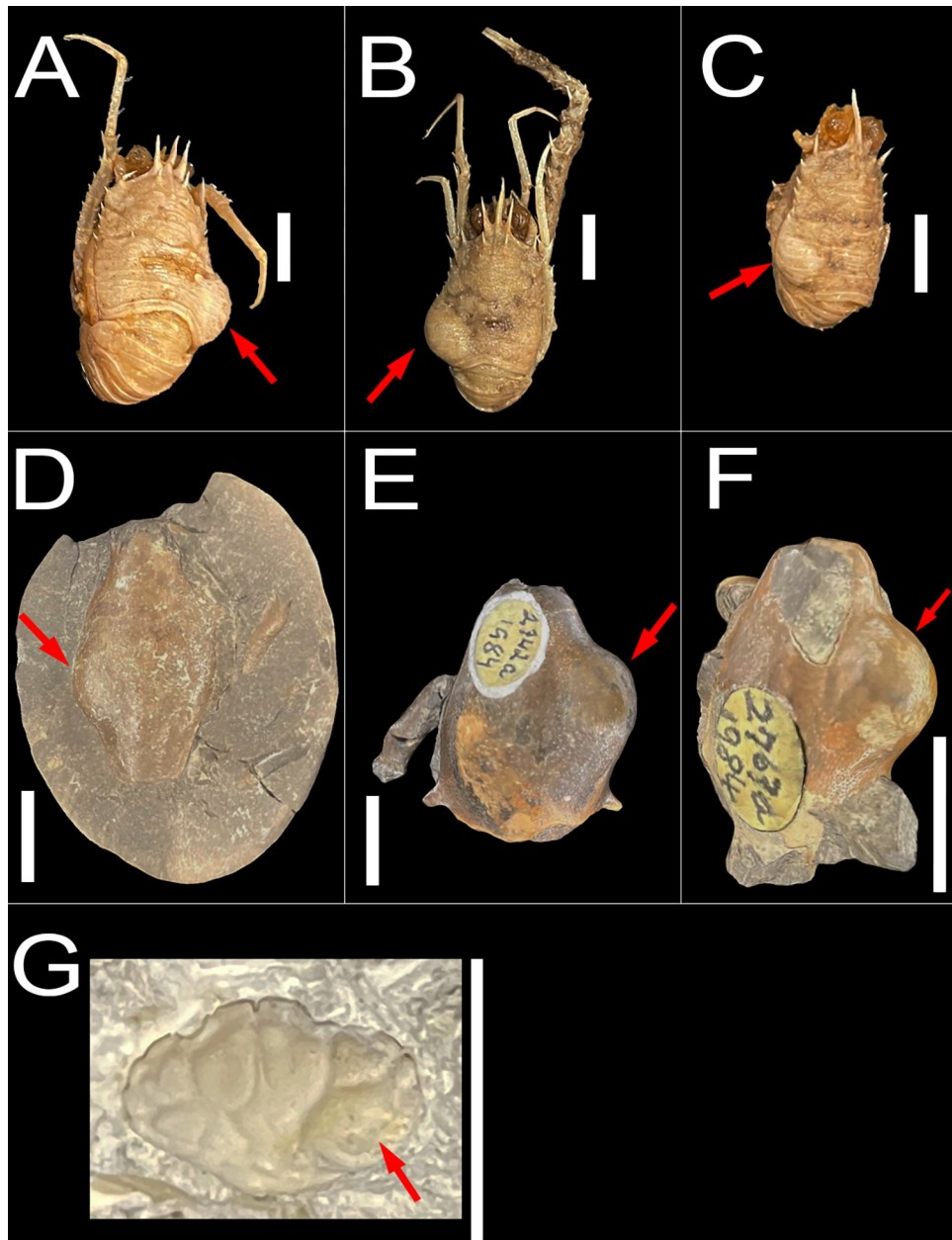

**Fig 1. Specimens with branchial swellings.** All scale bars are 1cm. Red arrows indicate swellings. A-C) Modern specimens of *Munida valida* TAMU cat. no. 2–3061 (A,C), 2–3063 (B). (D-F) Fossil specimens of *Macroacaena rosenkrantzi* NHMD MGUH 34322 (D), MGUH 34323 (E), MGUH 34324 (F). G) Fossil specimen of *Panopeus nanus* UF 288470.

Segmented specimen data was rendered for visualization in Dragonfly, then exported as Stereolithography (STL) 3D model files. Specimen STLs were imported into the software Blender 3.4 [43] for additional cleaning and processing, including removal of unconnected elements, remeshing, downscaling, and fixing geometry errors. After cleaning and fixing errors, each model was standardized to a uniform size and to a uniform number of faces (20,000±1,000) to improve consistency between specimens.

## Finite elements analysis

Finite elements analysis (FEA) is a simplified method of assessing the load and distribution of physical stresses on a complex 3D model with given material properties, forces, and constraints, by subdividing a complex model into finite parts, then solving partial differential equations for the parts individually [44]. The prepared specimen 3D models were imported into the software FreeCAD 0.20.2 [45] and converted into FEA models using the program Gmsh 4.11.1 [46]. In FreeCAD, identical material properties were applied to each model, which were amalgamated from multiple studies of the material properties of decapod skeletal elements [47–51]. The ventral surfaces of the specimens were set as constraints, and a force of ten newtons of point load force was applied at perpendicular angles to faces at the same location on both the left and right branchial chamber of each specimen (for each specimen, this included one normal branchial chamber and one swollen branchial chamber) (Fig 2). All FEA was also run using applied forces of five newtons and twenty newtons, to assess results sensitivity. Swanson et al. (2013) [52] demonstrated that ten newtons may be sufficient to indent or fracture a crustacean carapace of similar sizes to the specimens included here. Blue crabs, common scavengers and predators of small crustaceans along the east coast of the Americas, have

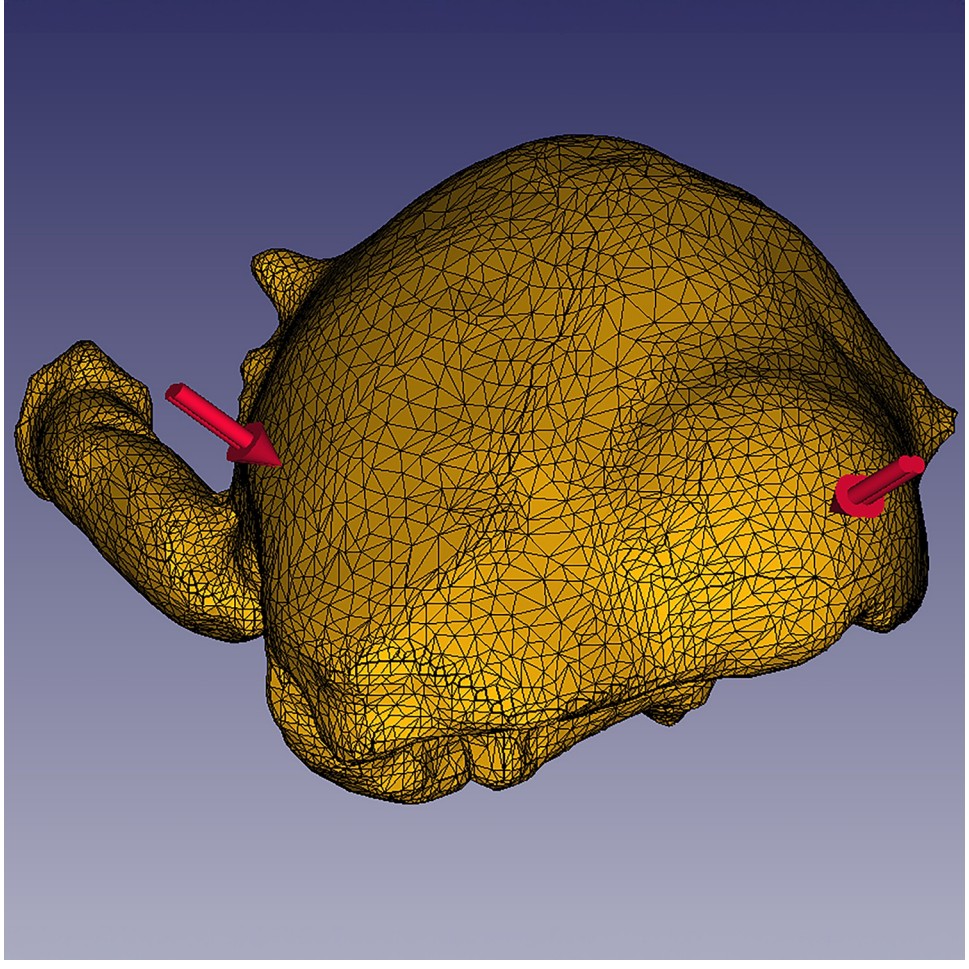

**Fig 2. Finite element model of *Macroacaena rosenkrantzi* with constraints in FreeCAD.** Red arrows indicate applied forces.

been shown to exert point load forces in excess of 10 newtons at the tip of the dactyl in vivo [53, 54]. The finite elements analysis was solved using the solver program CalculiX 2.10 [55], then the results were exported into Paraview 5.11.0 [56] as Visualization Toolkit (VTK) files for observation and visualization of FEA results. Von Mises stress, the stress response of a given material relative to the limit at which the material deforms, was chosen as the primary FEA output metric for characterizing the stress, strain, and deformation of the models in response to a force [57]. The vertex data from the left and right side of each mesh were isolated from each other to compare von Mises stress values between the normal and swollen side of each specimen.

## Results

### Computed tomography

Specimen CT scans output high-resolution 3D data, with a voxel size of 30.5 microns for the specimen of *P. nanus*, and a voxel size of 26.4 microns for the *Munida valida* and *Macroacaena rosenkrantzi* specimens (Fig 3A–3G). The higher resolution of the *Munida* and *Macroacaena* specimens is a result of the SubpiX scanning technique [41]. CT data of the recent *Munida* specimens revealed complex internal morphology and considerable heterogeneity of cuticle density and thickness (Fig 3A–3C). The scans of the fossil *Macroacaena* in concretions had no recognizable internal morphology, or internal evidence of the isopod parasite, although some

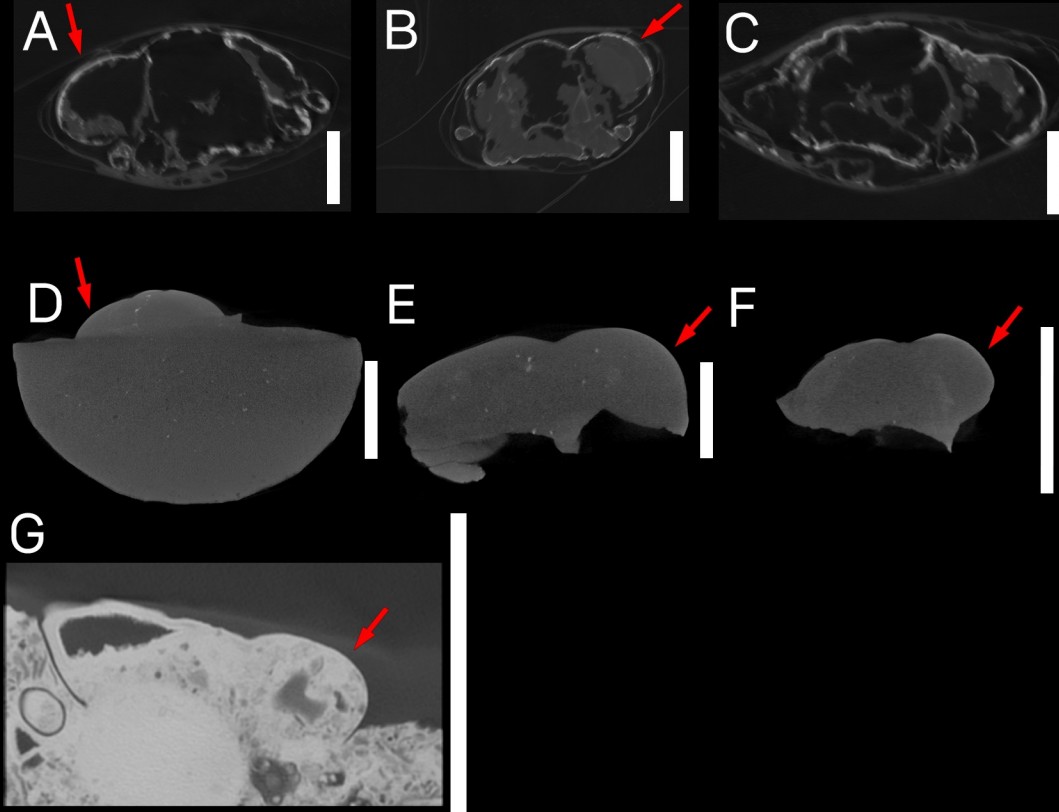

**Fig 3. CT data of specimens.** All specimens are shown in transverse cross-section. Scale bars are 1 cm, shown to the right of each specimen. Red arrows indicate swellings. A-C) Modern specimens of *Munida valida* TAMU cat. no. 2–3061 (A,C), 2–3063 (B). D-F) Fossil specimens of *Macroacaena rosenkrantzi* NHMD MGUH 34322 (D), MGUH 34323 (E), MGUH 34324 (F). G) Fossil specimen of *Panopeus nanus* UF 288470.

of the original host cuticle was intact and is distinguishable from the concretion (Fig 3D–3F). The *Panopeus* specimen CT data revealed that it is an isolated carapace, possibly a molt, with no preserved cuticle, and that the swollen right side of the carapace is filled with matrix, while the normal left side of the carapace contains a large void (Fig 3G).

## Finite elements analysis

The finite elements analysis output meshes comprised of tens of thousands of vertices which store individual stress, strain, and deformation values. The stress was highly localized to the area immediately around the site of applied force in all models and was not distributed to other areas of the branchial chamber or carapace (Fig 4). The peak stress on the swollen chamber was higher than on the healthy chamber for all specimens, although the magnitude of the difference varied considerably between specimens, from a 1.53% difference in the specimen of *P. nanus*, to a 51.00% difference in *M. valida* specimen A, with a mean of a 24.13% difference in peak stress, and median of 20.54% peak stress across all specimens (Figs 4 and 5, Table 1). A

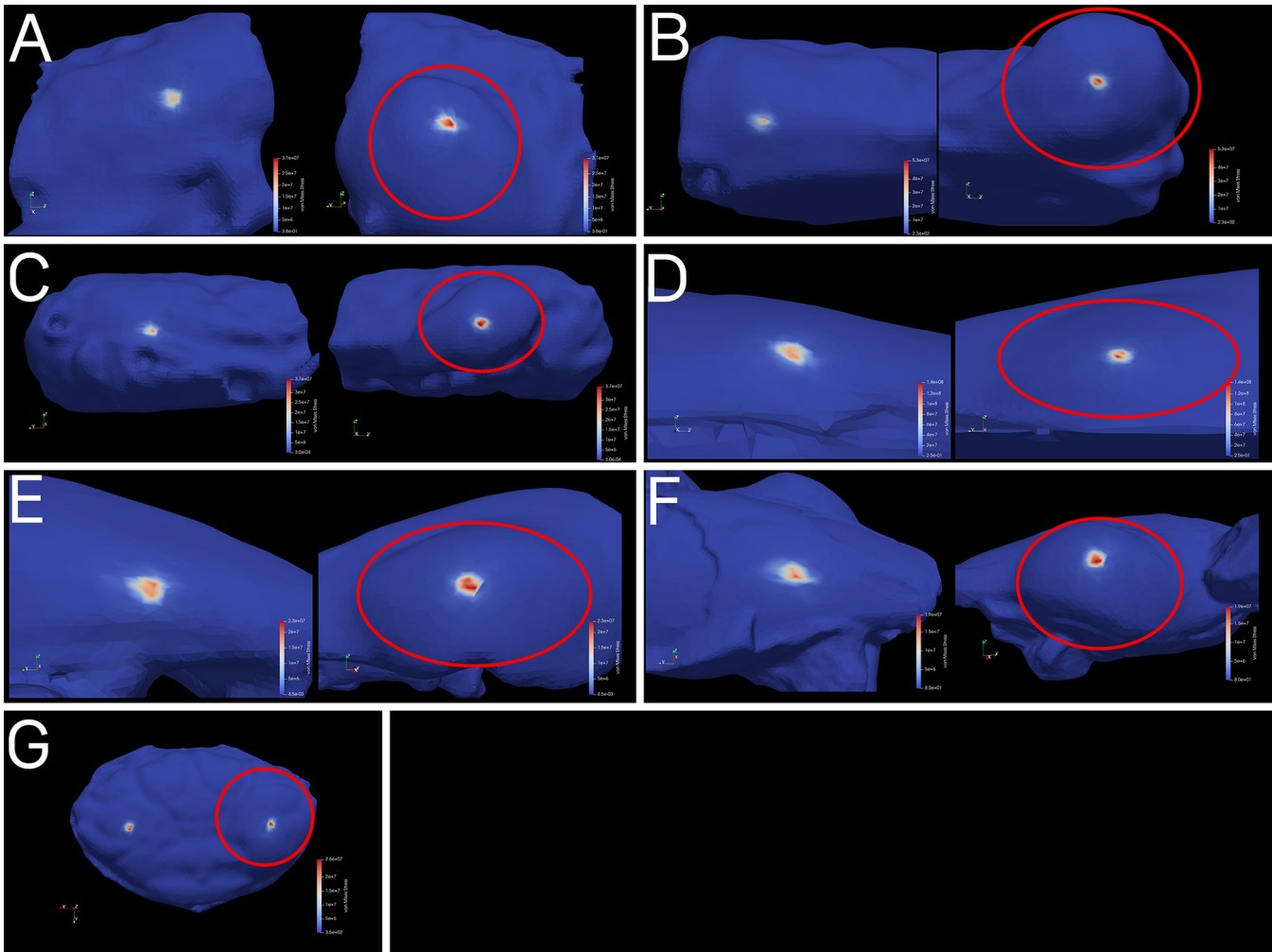

**Fig 4. FEA results meshes.** Swellings are displayed to the right and circled in red for each specimen. Values on scales are in pascals (Pa). A-C) Modern specimens of *Munida valida* TAMU cat. no. 2–3061 (A,C), 2–3063 (B). D-F) Fossil specimens of *Macroacaena rosenkrantzi* NHMD MGUH 34322 (D), MGUH 34323 (E), MGUH 34324 (F). G) Fossil specimen of *Panopeus nanus* UF 288470.

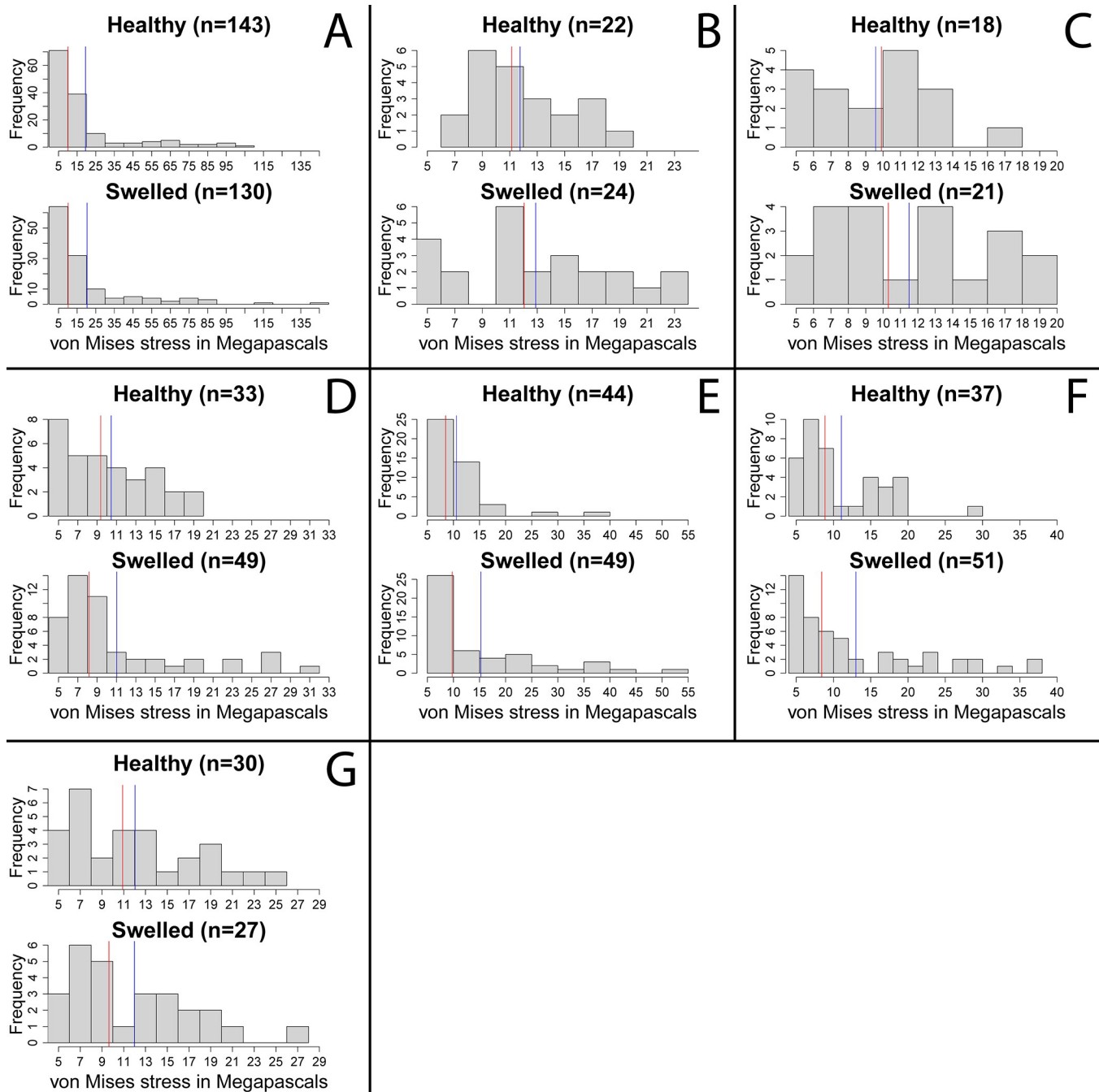

**Fig 5. Histograms of model vertex von Mises stress values exceeding 5 MPa.** Red lines indicate the median stress values. Blue lines indicate the mean stress values. The number of vertices exceeding 5MPa is given for each side of each specimen. A) *Munida valida* specimen A TAMU cat. no. 2–3061. B) *M. valida* specimen B TAMU cat. No. 2–3063. C) *M. valida* specimen C TAMU cat. no. 2–3061. D) *Macroacaena rosenkrantzi* specimen A NHMD MGUH 34322. E) *M. rosenkrantzi* specimen B NHMD MGUH 34323. F) *M. rosenkrantzi* specimen C NHMD MGUH 34324. G) *Panopeus nanus* UF 288470.

one-tailed paired Mann-Whitney test performed in R shows a statistically significant difference (p = 0.008) in the peak stress between the healthy and swollen chambers, although the broader frequency distribution of high (>5MPa) stress vertices across the surface of the models (Fig 5) are similar between healthy and swollen chambers. The broader trend of highly

**Table 1. Summary of von Mises stress values from specimen FEA.**

|  | Healthy—Peak Stress | Healthy—Mean Stress | Healthy—Mean Stress above 5MPa | Healthy—Median Stress above 5MPa | Percent Difference in Peak Stress | Swollen—Peak Stress | Swollen—Mean Stress | Swollen—Mean Stress above 5MPa | Swollen—Median Stress above 5MPa |
|---|---|---|---|---|---|---|---|---|---|
| *Munida valida* A TAMU cat. no. 2–3061 | 18.7 | 0.0924 | 10.45 | 9.37 | 51.00% | 31.5 | 0.1521 | 11.02 | 8.16 |
| *Munida valida* B TAMU cat. no. 2–3063 | 39.4 | 0.076 | 10.56 | 8.51 | 29.81% | 53.2 | 0.0877 | 15.24 | 9.75 |
| *Munida valida* C TAMU cat. no. 2–3061 | 29.7 | 0.0763 | 11.07 | 8.87 | 20.54% | 36.5 | 0.101 | 13.01 | 8.42 |
| *Macroacaena rosenkrantzi* A NHMD MGUH 34322 | 102 | 0.1574 | 19.52 | 10.1 | 32.79% | 142 | 0.1425 | 20.36 | 10.2 |
| *Macroacaena rosenkrantzi* B NHMD MGUH 34323 | 19.2 | 0.0947 | 11.75 | 11.15 | 20.14% | 23.5 | 0.0953 | 12.89 | 12.05 |
| *Macroacaena rosenkrantzi* C NHMD MGUH 34324 | 17.1 | 0.0721 | 9.57 | 9.90 | 13.11% | 19.5 | 0.0675 | 11.49 | 10.3 |
| *Panopeus nanus* UF 288470 | 25.9 | 0.0574 | 12.04 | 10.9 | 1.53% | 26.3 | 0.0516 | 11.99 | 9.64 |

All stress values are in megapascals (MPa).

localized stress peaks, resulting in right-skewed distributions, is observed in most specimens even when excluding smaller stress values. These distributions of stress and differences in peak stress are insensitive to changes in input force, except for the specimen of *P. nanus*, which had peak stress vary between the swollen and healthy chamber at different input forces but maintained peak stress differences below 2% (tested at 5N and 20N, S1 and S2 Figs, S1 Table). Differences in peak stress are also statistically significant for the other input forces (5N and 20N) using the same statistical test as the 10 newton peaks, with a p-value of 0.016 at both alternate forces. FEA results stress values are stored and accessible through the repository Dryad [58].

## Discussion

The results of the finite elements analysis for the seven specimens are indicative of higher stress at a swollen branchial chamber relative to a healthy branchial chamber resulting from the same force. Four of the specimens have greater overall mean stress on the swollen chamber relative to the healthy chamber, and all specimens except for the *M. rosenkrantzi* specimen C have greater median stress on the swollen chamber, although due to the highly localized nature of each models' stress response (Fig 4), and the greater number of vertices on a swollen branchial region resulting from the shape deformity relative to a normal branchial region, means and medians of stress values are not directly comparable. When the results include only the vertices with high stress values around the site of the applied force (above 5 MPa: Fig 5 and Table 1), the number of vertices is similar between healthy and swollen chambers, four of the specimens have greater median stress on the swollen chamber, and the mean stress is higher on the swollen chamber for all but the specimen of *P. nanus*. These patterns, as well as the overall distribution of stress and differences in peak stress, are insensitive to changes in the magnitude of applied force, although the higher peak stress on the specimen of *P. nanus* varied between the healthy and swollen chamber at different forces while maintaining a peak difference below 2% (S1 and S2 Figs, S1 Table). This difference in *P. nanus*, a small difference in peak stress between normal and swollen regions, and greater variability of results under

different forces, is likely a reflection of the considerably different body plan of *Panopeus* relative to *Munida* and *Macroacaena*. The relatively elongated and rounded body plans of *Munida* and *Macroacaena* resulted in laterally exaggerated branchial swellings relative to the flat and wide body plan of *Panopeus*, which has the least visually apparent swelling of the specimens studied. This difference may reflect broader differences in the impact of *K. crusta* on preservation across decapod taxa and body plans.

The observed trend, with greater peak stress in response to the same force at the site of a parasite-induced branchial swelling relative to a normal branchial chamber for each specimen, may represent an early indication of a shape-related decrease in fossil preservation potential for *Kanthyloma crusta*. We posit that greater stress in a swollen branchial chamber relative to a healthy branchial chamber in response to identical forces may result in a preservation bias against intact fossil preservation of the swelling and the host. Assuming that on average both branchial chambers in a specimen are experiencing external forces at similar magnitude and frequency, we would expect that the swollen branchial chamber would preserve less often than the healthy branchial chamber and would reach the point of fracture at lower magnitude and frequency of forces than the healthy chamber. For this reason, we also expect specimens with *K. crusta* to preserve intact less frequently than specimens without *K. crusta*. The magnitude of this preservation bias is not yet known, nor is it directly addressed herein, but the observed trend is a first step towards understanding the effect of taphonomy on the preservation of these parasitic traces. These results need to be supported by experimental evidence and other methods of taphonomy to better interpret FEA data and better understand the quantitative implications of our results.

If our suggestion is correct, a taphonomic bias against the preservation of this parasitic trace implies reduced fossil prevalence and host diversity relative to the fossil record of unparasitized hosts. These results emphasize the importance of recontextualizing the fossil record of *K. crusta* and parasitism more broadly to disentangle fossil record and sampling biases from true signals in the fossil record of parasitism. Non-destructive, computationally intensive methods, as presented here, are key to revealing ecological and evolutionary trends in the fossil record of parasitism. Resolving this gap is crucial as it has become imperative to understand and predict how a changing climate will impact parasite-host dynamics. Studies predicting changes in parasite-host dynamics relating to anthropogenic change have produced varying results, owing to geographic and taxonomic unevenness of climate change impacts. However, it appears likely that climate change will induce shifts in the range, prevalence, and diversity of parasites, leading to widespread consequences for global ecosystems [59–63]. Some studies modelling parasite climate vulnerability based on occurrence data or functional traits suggest that climate driven habitat loss and coextinction driven by host extinction are likely to cause widespread extinction in major parasitic clades within 100 years, although the impacts will be most severe for ectoparasites, parasites with narrow climate tolerance, and parasites with high host specificity [59, 60]. Parasites are important regulatory members of ecosystems [1], both increases and declines in the prevalence or geographic range of the parasites studied here will have consequences for their hosts and the broader ecosystem. Fossil parasite data could be used to set ecological baselines for parasite geographic range and prevalence, and could be used to study the response of parasite ecology to past intervals of rapid warming (i.e. the PETM) [64], but the known fossil record of parasitism is too sparse at present. With a thorough understanding of the potential biases that affect the fossil record of parasites we may be able to quantitatively use the fossil record of parasites towards meaningful outcomes for conservation of ecosystems and aquacultural resources.

Fossil preservation is a complex combination of biological, ecological, and environmental factors and conditions which are not fully explored here. Although point load forces as used in

this analysis can be applied by some common crustacean scavengers and predators [53, 54], point load forces do not represent the full range of forces and pressures that decapod skeletal material may be subjected to before burial. Decapods exhibit considerable differences in shape, cuticle thickness, and calcification between species, as well as ecological differences between taxa which have considerable taphonomic impacts [14]. In addition, the presence of soft tissue and reabsorption of calcium before molting may result in considerable differences in preservation between corpses and molts [65]. Each of these factors must be examined going forward to understand the diverse preservation characteristics of crustaceans and their parasites through time, and subsequently crustacean parasite-host dynamics throughout earth's history. In this study of seven specimens across three species, the changes to body shape induced by a crustacean parasite swelling were isolated from the many other factors affecting preservation. Thus, there is considerable work still to be done, both in understanding the relationship between shape and preservation in these fossil parasite traces, and in contextualizing the preservation of the fossil record of crustacean parasites more broadly. The methods presented here must be further refined and applied to better characterize the preservation of parasites in the marine fossil record. The complexity of preservation necessitates future refinements to the FEA methods used here, which should strive to incorporate increased model resolution, improved fidelity of material properties, and implementation of carapace thickness, as well as implementing advances to the fidelity and application of FEA in paleontology such as non-linearities [66]. The significant differences between species and individuals also emphasize the importance of investigating patterns of parasite-host preservation with larger numbers of fossil and modern specimens, and across a larger number of species. These computationally intensive scanning and analytical methods are especially valuable non-destructive tools for in-depth study of rare fossil occurrences, such as parasitic traces, but it is also critical to ground truth these results through experimentation with modern proxies, as well as thorough study of institutional fossil collections.

## Supporting information

**S1 Fig. FEA results meshes for 5N force.** Swellings are displayed to the right and circled in red for each specimen. A-C) Modern specimens of *Munida valida* TAMU cat. no. 2–3061 (A, C), 2–3063 (B). D-F) Fossil specimens of *Macroacaena rosenkrantzi* NHMD MGUH 34322 (D), MGUH 34323 (E), MGUH 34324 (F). G) Fossil specimen of *Panopeus nanus* UF 288470. (TIF)

**S2 Fig. FEA results meshes for 20N force.** Swellings are displayed to the right and circled in red for each specimen. A-C) Modern specimens of *Munida valida* TAMU cat. no. 2–3061 (A, C), 2–3063 (B). D-F) Fossil specimens of *Macroacaena rosenkrantzi* NHMD MGUH 34322 (D), MGUH 34323 (E), MGUH 34324 (F). G) Fossil specimen of *Panopeus nanus* UF 288470. (TIF)

**S1 Table. Summary of von Mises stress values from specimen FEA at 5N and 20N.** All stress values are in Megapascals (MPa).
(XLSX)

## Acknowledgments

We thank the following individuals and institutions for their guidance and kind loans of the specimens used in this study: Mary Wicksten and Texas A&M University; Roger Portell and the University of Florida, Florida Museum of Natural History, Invertebrate Paleontology;

Laura Cotton, Arden Bashforth, and the Natural History Museum of Denmark. We thank Murtada Naser, Carrie Schweitzer and Daniel Lima for their constructive reviews.

## Author Contributions

**Conceptualization:** Nathan L. Wright, Elizabeth Petsios.

**Data curation:** Nathan L. Wright.

**Formal analysis:** Nathan L. Wright.

**Investigation:** Nathan L. Wright.

**Methodology:** Nathan L. Wright.

**Project administration:** Elizabeth Petsios.

**Resources:** Elizabeth Petsios.

**Supervision:** Elizabeth Petsios.

**Visualization:** Nathan L. Wright.

**Writing – original draft:** Nathan L. Wright.

**Writing – review & editing:** Nathan L. Wright, Adiël A. Klompmaker, Elizabeth Petsios.

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
