## [Decision Letter · Decision Letter 0]

29 Jan 2024

PONE-D-23-40807Exploring the preservation of a parasitic trace in decapod crustaceans using finite elements analysisPLOS ONE

Dear Dr. Wright,

Thank you for submitting your manuscript to PLOS ONE. After careful consideration, we feel that it has merit but does not fully meet PLOS ONE’s publication criteria as it currently stands. Therefore, we invite you to submit a revised version of the manuscript that addresses the points raised during the review process.

In general, the Manuscript is skillfully written and concise, introducing a novel way for examining the preservation of parasite evidence in crustaceans. This methodology has just been utilized in the field of paleontology in recent times, signifying the initial endeavor to employ it for this specific objective. Hence, the manuscript presents unique data and delineates the preliminary stages of Finite Element Analysis (FEA) for investigating parasitism in crustaceans.

The manuscript has been reviewed by two reviewers and based on these revisions, the authors need to edit or correct their manuscript accordingly.

We look forward to receiving your revised manuscript.

Kind regards,

Murtada D. Naser

Academic Editor

PLOS ONE

2. In your manuscript, please provide additional information regarding the specimens used in your study. Ensure that you have reported human remain specimen numbers and complete repository information, including museum name and geographic location.

For more information on PLOS ONE's requirements for paleontology and archeology research, see https://journals.plos.org/plosone/s/submission-guidelines#loc-paleontology-and-archaeology-research.

4. We note that Figure 1 in your submission contain copyrighted images. All PLOS content is published under the Creative Commons Attribution License (CC BY 4.0), which means that the manuscript, images, and Supporting Information files will be freely available online, and any third party is permitted to access, download, copy, distribute, and use these materials in any way, even commercially, with proper attribution. For more information, see our copyright guidelines: http://journals.plos.org/plosone/s/licenses-and-copyright.

5. We notice that your supplementary tables are included in the manuscript file. Please remove them and upload them with the file type 'Supporting Information'. Please ensure that each Supporting Information file has a legend listed in the manuscript after the references list.

Reviewers' comments:

Reviewer's Responses to Questions

**Comments to the Author**

1. Is the manuscript technically sound, and do the data support the conclusions?

Reviewer #1: Yes

Reviewer #2: Partly

2. Has the statistical analysis been performed appropriately and rigorously? 

Reviewer #1: I Don't Know

Reviewer #2: Yes

3. Have the authors made all data underlying the findings in their manuscript fully available?

Reviewer #1: Yes

Reviewer #2: Yes

4. Is the manuscript presented in an intelligible fashion and written in standard English?

Reviewer #1: Yes

Reviewer #2: Yes

5. Review Comments to the Author

Reviewer #1: I have read the paper by Wright et al. on parasitic traces in decapod crustaceans.

The paper is original, and I have not seen the results published elsewhere. The analyses with CT and FEA appear to have been conducted using appropriate and typical protocols. The paper is well-written with just a few suggestions to grammar and sentence structure that to me would make some sentences more clear. Data are designated for sharing in appropriate venues (Dryad).

I have a few questions about the results and the interpretation of the data.

1. Table 1 and Figure 5 both show that in many cases, there is not much difference between stress on healthy and swollen branchial regions, and lines 185-186 state “minimal differences in distribution between the normal and swollen…” Thus, I am wondering if the differences are statistically significant and what that would mean? Are the normal sides significantly less stressed than the swollen sides?

2. The authors state that the stress might induce a shape-related decrease in preservation potential (lines 235-238). How and why? Does the stressed region break apart more easily? Is the cuticle thinned by the stress? If that is true, do we have evidence of broken bopyrid swellings or carapaces that have a branchial region missing? Or does the swelling change the hydrodynamics of the carapace? The relationship between the stress and preservation potential needs to be specifically enumerated.

3. Line 244 suggests that it is important to understand and predict how parasite-host dynamics will change with climate change. Why? Can important outcomes be documented?

4. Lines 245-249 mention studies of anthropogenic change and its impact on parasite-host dynamics. Can these be described?

5. Pursuant to 3 and 4 above, how will the study of the stress on the branchial chambers help to resolve these questions?

Addressing these questions would make the paper very strong.

Reviewer #2: Overall, the manuscript is well-written and concise, presenting a new methodology for investigating the preservation of parasitic traces in crustaceans. This approach has only been employed in paleontology recently, marking the first attempt to use it for this purpose. Consequently, the manuscript provides original data and outlines the initial steps in Finite Element Analysis (FEA) for exploring parasitism in crustaceans.

All the data have been deposited in appropriate repositories. However, there is a need for more detailed descriptions of the methodology to facilitate the reproducibility of the experiments. For instance, specific details regarding the CT set parameters, such as voltage, current, exposure time per scan, voxel size resolution, and total scan, should be provided.

I acknowledge the importance of developing new methodologies and exploring alternative approaches to enhance the study and assessment of the fossil record of parasitism. However, I remain somewhat skeptical about how FEA can contribute to understanding the impact of carapace shape deformations on the physical preservation potential of the host.

While I am not an expert in FEA, I believe it is crucial that the manuscript is easily comprehensible to non-specialists. Therefore, I recommend providing a clear and accessible explanation of the methodology, as well as presenting its results and discussions in a manner that is easily understandable for readers who may not have a background in this specific field.

6. PLOS authors have the option to publish the peer review history of their article (what does this mean?). If published, this will include your full peer review and any attached files.

Reviewer #1: **Yes: **Carrie E Schweitzer

Reviewer #2: **Yes: **Daniel Lima

---

## [Author Response · Author response to Decision Letter 0]

13 Mar 2024

Our responses to each comment begin with an asterisk. All line numbers refer to the marked-up manuscript “Revised Manuscript with Track Changes”. These responses are also included in the uploaded file "Response to Reviewers".

Journal/Editor comments and requirements

comment 1) Please ensure that your manuscript meets PLOS ONE's style requirements, including those for file naming. 

*We have adjusted lines 11-12 to add a corresponding author, lines 174 and 192 to change the level 2 headings to sentence case, and lines 533-537 to correct the positioning of the legend for S1 table. 

comment 2) In your manuscript, please provide additional information regarding the specimens used in your study. Ensure that you have reported human remain specimen numbers and complete repository information, including museum name and geographic location.

*Geographic locations of the museum collections and specimens were added in lines 106-112. No permits were required for this study, and all relevant regulations were followed, and the statement reflecting this was added to lines 116-117. Should this manuscript be accepted, the Natural History Museum of Denmark intends to issue new specimen numbers for their loaned specimens used in this study (which are currently NHMD GM 1985.886, GM 1984.2742, GM 1984.2763), which will need to be reflected in the published version of the manuscript if it is accepted.

comment 3) When completing the data availability statement of the submission form, you indicated that you will make your data available on acceptance. We strongly recommend all authors decide on a data sharing plan before acceptance, as the process can be lengthy and hold up publication timelines. Please note that, though access restrictions are acceptable now, your entire data will need to be made freely accessible if your manuscript is accepted for publication. This policy applies to all data except where public deposition would breach compliance with the protocol approved by your research ethics board. If you are unable to adhere to our open data policy, please kindly revise your statement to explain your reasoning and we will seek the editor's input on an exemption. Please be assured that, once you have provided your new statement, the assessment of your exemption will not hold up the peer review process.

*The FEA stress data has been prepared in Dryad and will be submitted and made available as soon as possible should the manuscript be accepted. One reason for this delay is that the Natural History Museum of Denmark intends to issue new specimen numbers for their loaned specimens used in this study, pending manuscript acceptance.

comment 4) We note that Figure 1 in your submission contain copyrighted images. All PLOS content is published under the Creative Commons Attribution License (CC BY 4.0), which means that the manuscript, images, and Supporting Information files will be freely available online, and any third party is permitted to access, download, copy, distribute, and use these materials in any way, even commercially, with proper attribution. For more information, see our copyright guidelines: http://journals.plos.org/plosone/s/licenses-and-copyright.

We require you to either (1) present written permission from the copyright holder to publish these figures specifically under the CC BY 4.0 license, or (2) remove the figures from your submission…

*Figure 1 contains images of the specimens using digital photography and digital reconstructions of multiple photographs using the software Polycam. All photographs, models, and input data for models were created by author Nathan Wright, who consents to publishing the images under the CC BY 4.0 license. Models and images of models created in Polycam are owned and publishable by the user that created them pursuant to Polycam terms and conditions sections 5.2 and 6.4. These multi-photo reconstructions were more suitable for figuring due to better angle, lighting, and scale than the uncombined photographs, and the loaned specimens have since been returned to the collections of the Natural History Museum of Denmark. A reference to Polycam was added in lines 117-119 to make it clearer that photogrammetry models were used for visualization, and specifically for figure 1 D-F.

comment 5) We notice that your supplementary tables are included in the manuscript file. Please remove them and upload them with the file type 'Supporting Information'. Please ensure that each Supporting Information file has a legend listed in the manuscript after the references list.

*We have removed S1 table from the manuscript file, uploaded as a separate supporting information file 

Reviewer #1 comments

I have read the paper by Wright et al. on parasitic traces in decapod crustaceans.

The paper is original, and I have not seen the results published elsewhere. The analyses with CT and FEA appear to have been conducted using appropriate and typical protocols. The paper is well-written with just a few suggestions to grammar and sentence structure that to me would make some sentences more clear. Data are designated for sharing in appropriate venues (Dryad).

comment 1) Table 1 and Figure 5 both show that in many cases, there is not much difference between stress on healthy and swollen branchial regions, and lines 185-186 state “minimal differences in distribution between the normal and swollen…” Thus, I am wondering if the differences are statistically significant and what that would mean? Are the normal sides significantly less stressed than the swollen sides?

*We thank the reviewer for highlighting the importance of statistical significance, and for pointing out some wording which was unnecessarily vague. We now have added statistical test to support our reasoning. There are statistically significant differences in peak stress (p = 0.007813) between healthy and swollen sides shown by a paired Mann-Whitney test, which is maintained in the tests with different input forces (p = 0.01563). The section which states minimal differences is referring specifically to the differences in the frequency distribution of model vertex stress values seen in the histograms of Figure 5. The sentence has been changed to better reflect our meaning. 

Lines 205-211: “A one-tailed paired Mann-Whitney test performed in R shows a statistically significant difference (p=0.008) in the peak stress between the healthy and swollen chambers, although the broader frequency distribution of high (>5MPa) stress vertices across the surface of the models (Fig 5) are similar between healthy and swollen chambers. The broader trend of highly localized stress peaks, resulting in right-skewed distributions, is observed in most specimens even when excluding smaller stress values.”

Lines 214-216: “Differences in peak stress are also statistically significant for the other input forces (5N and 20N) using the same statistical test as the 10 newton peaks, with a p-value of 0.016 at both alternate forces.”

comment 2) The authors state that the stress might induce a shape-related decrease in preservation potential (lines 235-238). How and why? Does the stressed region break apart more easily? Is the cuticle thinned by the stress? If that is true, do we have evidence of broken bopyrid swellings or carapaces that have a branchial region missing? Or does the swelling change the hydrodynamics of the carapace? The relationship between the stress and preservation potential needs to be specifically enumerated.

*We have strengthened our discussion on our interpretation of the FEA results and the connection to preservation in lines 262-274: “We posit that greater stress in a swollen branchial chamber relative to a healthy branchial chamber in response to identical forces may result in a preservation bias against intact fossil preservation of the swelling, and the host. Assuming that on average both branchial chambers in a specimen are experiencing external forces at similar magnitude and frequency, we would expect that the swollen branchial chamber would preserve less often than the healthy branchial chamber and would reach the point of fracture at lower magnitude and frequency of forces than the healthy chamber. For this reason, we also expect specimens with K. crusta to preserve intact less frequently than specimens without K. crusta. The magnitude of this preservation bias is not yet known, nor is it directly addressed herein, but the observed trend is a first step towards understanding the effect of taphonomy on the preservation of these parasitic traces. These results need to be supported by experimental evidence and other methods of taphonomy to better interpret FEA data and better understand the quantitative implications of our results.”

While we have isolated a single aspect (shape) from the highly complex set of interconnecting factors that influence fossil preservation, we assert that a higher stress response in reaction to identical forces suggests that a swollen branchial chamber is more likely to fracture and less likely to preserve intact than a healthy branchial chamber, assuming both chambers experience external forces at similar frequency and magnitude. This suggests that the swelling itself is less likely to preserve intact than the healthy branchial chamber, but also that a specimen with this swelling is less likely to preserve intact than a specimen without the swelling. Remaining diagnostically intact is especially important for traces on fossils, as shape is a key factor in identifying Kanthyloma crusta (and many other traces of biotic interactions) relative to other possible abnormalities or diagenetic changes.

Some questions the reviewer asked are not easily answered. 

Robins et al. (2016: Ann. Naturhist. Mus. Wien, Serie A): “Holotype of Catillogalathea falcula nov. spec. (NHMW 2007z0149/0435a) from the Ernstbrunn Limestone.” Example of a swelling that is broken in part, prior to burial I think given that the matrix is present, but rest of carapace is there. One of the rare examples I could find among many swollen specimens. We could mention it as a rare example that specimens with swelling can break there prior to burial and preserve. What it does mean for taphonomy is another matter.

I found one example (NHMW 1990/0041/0368; Tithonian, Ernstbrunn) where just the swollen part of a crab was preserved and a bit of the surrounding cuticle.

Although this study is about shape, one aspect to consider too is cuticle thickness. Bursey (1978) showed, for one species, that the calcified cuticle thickness did not vary between a healthy and swollen branchial region. If this applies to other species too, cuticle thickness does not to be taken into account. Bursey CR (1978) Histopathology of the parasitization of Munida iris (Decapoda: Galatheidae) by Munidion irritans (Isopoda: Bopyridae). Bull Mar Sci 28:566–570

comment 3) Line 244 suggests that it is important to understand and predict how parasite-host dynamics will change with climate change. Why? Can important outcomes be documented?

*We have expanded our discussion on the importance of understanding parasitism in the fossil record in relation to future change in lines 291-296: “Parasites are important regulatory members of ecosystems [1], both increases and declines in the prevalence or geographic range of the parasites studied here will have consequences for their hosts and the broader ecosystem. Fossil parasite data could be used to set ecological baselines for parasite range and prevalence, and could be used to study the response of parasite ecology to past intervals of rapid warming (i.e. the PETM) [64], but the known fossil record of parasitism is too sparse at present.”

comment 4) Lines 245-249 mention studies of anthropogenic change and its impact on parasite-host dynamics. Can these be described?

*We have gone into greater detail on the literature of climate change’s impacts on parasitism in lines 286-291: “Some studies, modelling parasite climate vulnerability based on occurrence data or functional traits, suggest that climate driven habitat loss and coextinction driven by host extinction are likely to cause widespread extinction in major parasitic clades within 100 years, although the impacts will be most severe for ectoparasites, parasites with narrow climate tolerance, and parasites with high host specificity [59-60].”

5) Pursuant to 3 and 4 above, how will the study of the stress on the branchial chambers help to resolve these questions?

*With a more thorough understanding of the taphonomic biases that affect the fossil record of parasites we may be able to use the fossil record of parasites in a quantitatively useful way for conservation of ecosystems and resources. Although we have not studied the exact magnitude of this bias for K. crusta, we plan to continue using FEA in conjunction with other methods to ultimately estimate the magnitude of preservation bias for a given parasitic trace fossil (or parasite body fossil). Understanding the bias in quantitative terms that can be applied to fossil data will allow us to better constrain parasite ecological signals in the fossil record to allow for important paleobiological outcomes such as creation of ecological baselines and understanding parasite responses to climate change. We have made this connection clearer:

Lines 264-269: “Assuming that on average both branchial chambers in a specimen are experiencing external forces at similar magnitude and frequency, we would expect that the swollen branchial chamber would preserve less often than the healthy branchial chamber and would reach the point of fracture at lower magnitude and frequency of forces than the healthy chamber. For this reason, we also expect specimens with K. crusta to preserve intact less frequently than specimens without K. crusta.”

Lines 293-299: “Fossil parasite data could be used to set ecological baselines for parasite geographic range and prevalence, and could be used to study the response of parasite ecology to past intervals of rapid warming (i.e. the PETM) [64], but the known fossil record of parasitism is too sparse at present. With a thorough understanding of the potential biases that affect the fossil record of parasites we may be able to quantitatively use the fossil record of parasites towards meaningful outcomes for conservation of ecosystems and aquacultural resources.”

Reviewer #2 comments

Overall, the manuscript is well-written and concise, presenting a new methodology for investigating the preservation of parasitic traces in crustaceans. This approach has only been employed in paleontology recently, marking the first attempt to use it for this purpose. Consequently, the manuscript provides original data and outlines the initial steps in Finite Element Analysis (FEA) for exploring parasitism in crustaceans.

comment 1) All the data have been deposited in appropriate repositories. However, there is a need for more detailed descriptions of the methodology to facilitate the reproducibility of the experiments. For instance, specific details regarding the CT set parameters, such as voltage, current, exposure time per scan, voxel size resolution, and total scan, should be provided.

*To make the methods clearer and more exact for reproducibility, we have included more details on the tools used and the CT parameters:

Lines 126-131: “…using a voltage of 80 kV, current of 480µA, with the specimen platform rotating at 4.5 degrees per second, a detector framerate of 12.5 fps, an average source to specimen distance of 128 mm, and a source to detector distance of 539 mm."

Lines 136-141: “…using Dragonfly’s ROI (region of interest) painter tools and manually trained AI-assistedmachine learning segmentation models created in Dragonfly’s Segmentation Wizard tool [42].”

comment 2) I acknowledge the importance of developing new methodologies and exploring alternative approaches to enhance the study and assessment of the fossil record of parasitism. However, I remain somewhat skeptical about how FEA can contribute to understanding the impact of carapace shape deformations on the physical preservation potential of the host.

*We have made the connection between stress/force response and preservation clearer in lines 262-274: “We posit that greater stress in a swollen branchial chamber relative to a healthy branchial chamber in response to identical forces may result in a preservation bias against intact fossil preservation of the swelling, and the host. Assuming that on average both branchial chambers in a specimen are experiencing external forces at similar magnitude and frequency, we would expect that the swollen branchial chamber would preserve less often than the healthy branchial chamber and would reach the point of fracture at lower magnitude and frequency of forces than the healthy chamber. For this reason, we also expect specimens with K. crusta to preserve intact less frequently than specimens without K. crusta. The magnitude of this preservation bias is not yet known, nor is it directly addressed herein, but the observed trend is a first step towards understanding the effect of taphonomy on the preservation of these parasitic traces. These results need to be supported by experimental evidence and other methods of taphonomy to better interpret FEA data and better understand the quantitative implications of our results.”

See also our response to reviewer 1’s second comment.

comment 3) While I am not an expert in FEA, I believe it is crucial that the manuscript is easily comprehensible to non-specialists. Therefore, I recommend providing a clear and accessible explanation of the methodology, as well as presenting its results and discussions in a manner that is easily understandable for readers who may not have a background in this specific field.

*We thank the reviewer for noting the importance of facilitating the accessibility of the methods and results to a wider audience, and have made changes to lines 126, 133-134, and 138-141 to make them clearer.

Additional changes 

*To improve clarity, grammar, and sentence structure, minor changes have been made to lines 38, 41, 46, 49, 60, 63, 93-95, 102, 126, 135-136, 138, 142, 146, 148, 164, 185-186, 237, 249, 251-252, 275, 313, 316, 320, 327-328

*The reviewers and editor have been added to the acknowledgements for their constructive comments. (lines 338-339).

*Changes were made to lines 450, 453, 456-457, and 486-487 to make the citation of software tools more consistent.

---

## [Editor Report · Decision Letter 1]

24 Mar 2024

Exploring the preservation of a parasitic trace in decapod crustaceans using finite elements analysis

PONE-D-23-40807R1

Dear Dr.Nathan Lloyd Wright,

We’re pleased to inform you that your manuscript has been judged scientifically suitable for publication and will be formally accepted for publication once it meets all outstanding technical requirements.

Kind regards,

Murtada D. Naser

Academic Editor

PLOS ONE
---

## [Editor Report · Acceptance letter]

4 Apr 2024

PONE-D-23-40807R1 

PLOS ONE

Dear Dr. Wright, 

I'm pleased to inform you that your manuscript has been deemed suitable for publication in PLOS ONE. Congratulations! Your manuscript is now being handed over to our production team.

Kind regards, 

on behalf of

Dr. Murtada D. Naser 

Academic Editor

PLOS ONE